# Safety-Enhanced Autonomous Driving Using Interpretable Sensor Fusion Transformer

Hao Shao[1][*]  Letian Wang[2][*]  Ruobing Chen[1]
Hongsheng Li[3]    Yu Liu[1,4][†]

[1]SenseTime Research      [2]University of Toronto      [3] The Chinese University of Hong Kong
[4] Shanghai Artificial Intelligence Laboratory

**Abstract:** Large-scale deployment of autonomous vehicles has been continually delayed due to safety concerns. On the one hand, comprehensive scene understanding is indispensable, a lack of which would result in vulnerability in rare but complex traffic situations, such as the sudden emergence of unknown objects. However, reasoning from a global context requires access to sensors of multiple types and adequate fusion of these multi-modal sensor signals, which is difficult to achieve. On the other hand, the lack of interpretability in learning models also hampers the safety with unverifiable failure causes. In this paper, we propose a safety-enhanced autonomous driving framework, named Interpretable Sensor Fusion Transformer (InterFuser), to fully process and fuse information from multi-modal multi-view sensors for comprehensive scene understanding and adversarial event detection. Besides, intermediate interpretable features are generated from our framework, which provide more semantics and are exploited to better constrain actions to be within the safe sets. We conducted extensive experiments on CARLA benchmarks, where our model outperforms prior methods, ranking **#1** on the public CARLA Leaderboard. Code is made available to facilitate further research.

**Keywords:** Autonomous driving, sensor fusion, transformer, safety

## 1  Introduction

Recently, rapid progress has been witnessed in the field of autonomous driving, while the scalable and practical deployment of autonomous vehicles on public roads is still far from feasible. Their incompetence is mainly observed in high-traffic-density scenes, where a large number of obstacles and dynamic objects are involved in the decision making. In these cases, currently deployed systems could exhibit incorrect or unexpected behaviours leading to catastrophic accidents [1, 2]. While many factors contribute to such safety concerns, two of the major challenges are: 1) how to recognize rare adverse events of long-tail distributions, such as the sudden emergence of pedestrians from road sides, and vehicles running a red light, which require a better understanding of the scenes with multi-modal multi-view sensor inputs; 2) how to verify the decision-making process, in other words, identify functioning/malfunctioning conditions of the system and the causes for failures, which requires interpretability of the decision-making system.

Safe and reliable driving necessitates comprehensive scene understanding. However, a single sensor generally cannot provide adequate information for perceiving the driving scenes. Single-image approaches can hardly capture the surrounding environment from multiple perspectives and cannot provide 3D measurements of the scene, while single-LiDAR approaches cannot capture semantic information such as traffic lights. Though there are existing works fusing information from multiple sensors, they either match geometric features between image space and LiDAR projection space by locality assumption [3, 4], or simply concatenate multi-sensor features [5, 6]. The interactions and relationships between multi-modal features are seldomly modeled, such as the interactions between

---

[*]Equal contribution
[†]Corresponding author

6th Conference on Robot Learning (CoRL 2022), Auckland, New Zealand.

multiple dynamic agents and traffic lights, or features in different views and modalities. To encourage reasoning with a global context, the attention mechanism of Transformer [7] is utilized. The recent TransFuser [8] adopts internal feature propagation and aggregation via a multi-stage CNN-transformer architecture to fuse bi-modal inputs, which harms sensor scalability and is limited to fusion between LiDAR and a single-view image. In this paper, we take a one-stage architecture to effectively fuse information from multi-modal multi-view sensors and achieve significant improvement. As shown in Fig. 1, we consider LiDAR input and multi-view images (left, front, right, and focus) as complementary inputs to achieve comprehensive full-scene understanding.

On the other hand, existing end-to-end driving methods barely have a safety ensurance mechanism due to the lack of interpretability of how the control signal is generated. To tackle such a challenge, there are efforts to verify the functioning conditions of neural networks instead of directly understanding the models [9, 10, 11]. Though being helpful for choosing different models for different conditions, these methods still lack feedback from the failure causes for further improvement. Inspired by humans' information collecting process [12], in addition to generating actions, our model also outputs intermediate interpretable features, which we call *safety mind map*. As shown in Fig. 1, the safety mind map provides information on surrounding objects and traffic signs. Unveiling the process of perceiving and decision-making, our model is improvable with clear failure conditions and causes. Moreover, by exploiting this intermediate interpretable information as a safety constraint heuristic, we can constrain the actions within a safe action set to further enhance driving safety.

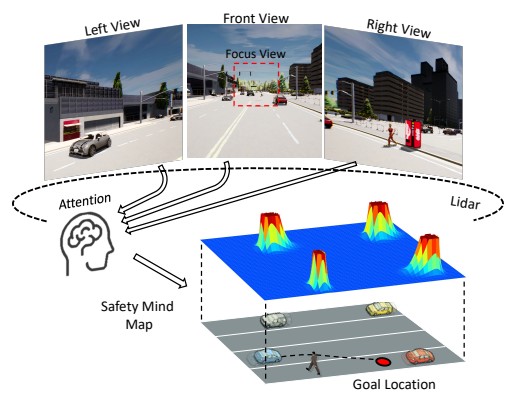

Figure 1: Safe and efficient driving requires comprehensive scene understanding by fusing information from multiple sensors. Peeking into the intermediate interpretable features of learning models can also unveil the model's decision basis. Such features enable improvable systems with access to failure causes, and can be used as safety heuristic to constrain actions within the safe set.

In this paper, we propose a safety-enhanced driving framework called Interpretable Sensor Fusion Transformer (InterFuser), in which information from multi-modal multi-view sensors is fused, and driving safety is enhanced by providing intermediate interpretable features as safety constraint heuristics. Our contributions are three-fold:

1. We propose a novel Interpretable Sensor Fusion Transformer (InterFuser) to encourage global contextual perception and reasoning in different modalities and views.

2. Our framework enhances the safety and interpretability of end-to-end driving by outputting intermediate features of the model and constraining actions within safe sets.

3. We experimentally validated our method on several CARLA benchmarks with complex and adversarial urban scenarios. Our model outperformed all prior methods, ranking the first on the public CARLA Leaderboard.

## 2    Related work

**End-to-end autonomous driving in urban scenarios**  The research of end-to-end autonomous driving based on the simulator of urban scenarios has become more and more popular. The topic starts from the development of an urban driving simulator: CARLA [13], together with Conditional Imitation Learning (CIL) [3]. A series of work follow this way, yielding conditional end-to-end driving [14, 15] in urban scenarios. LBC [16] proposed the mimicking methods aimed at training image-input networks with the supervision of a privileged model or squeezed model. Chitta et al. [17] presents neural attention fields to enable the reasoning for end-to-end driving models. Imitation learning(IL) approaches lack interpretability and their performance is limited by their handcrafted expert autopilot. Hence, researchers develop Reinforcement Learning (RL) agents to interact with simulated environments. Latent DRL [18] generates intermediate feature embedding from a top-

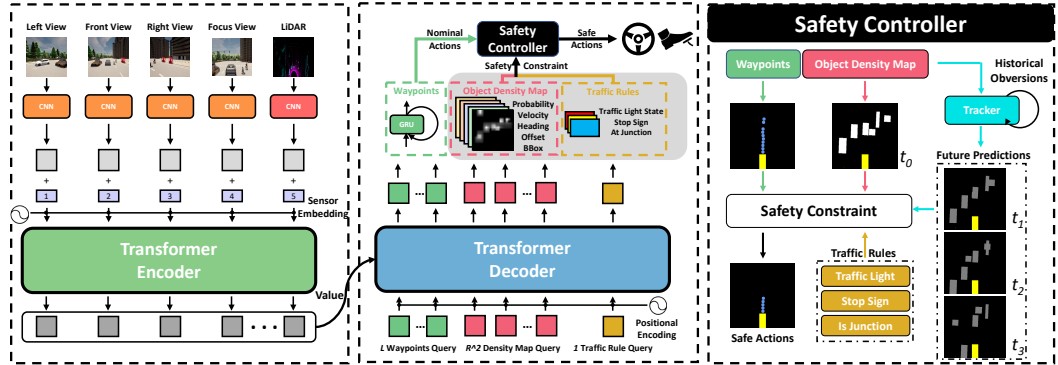

Figure 2: Overview of our approach. We first use CNN backbones to extract features from multi-modal multi-view sensor inputs. The tokens from different sensors are then fused in the transformer encoder. Three types of queries are then fed into the transformer decoder to predict waypoints, object density maps and traffic rules respectively. At last, by recovering the traffic scene from the predicted object density map and utilizing the tracker to forecast the future motion of other objects, a safety controller is applied to enhance the safety and efficiency of driving in complex traffic situations.

down view image by training a variational auto-encoder. With the aforementioned mimicking tricks, Roach [19] trained an RL-based privileged model as the expert agent to provide demonstrations for the IL agent. Toromanoff et al. [6] proposes to use hidden states which is supervised by semantic information as the input of RL policy.

**Transformer model in vision comprehension** Transformer was originally established in Natural Language Processing (NLP) [7]. The attention mechanism demonstrates to be a powerful module in image processing tasks. Vision Transformer (ViT) [20] computes relationships among pixel blocks with a reasonable computation cost. Later generations move on to generalize Transformer to other computer vision tasks [21, 22, 23, 24] or digging deeper to perform better [25, 26]. The attention mechanism brings a new entry point for modality fusion. TransformerFusion [27] reconstruct 3D scenes that takes monocular video as input with a transformer architecture. TransFuser [8] exploits several transformer modules for the fusion of intermediate features of front view and LiDAR. However, such a sensor-pair intense fusion approach hardly scales to more sensors, while information from side views like an emerging pedestrian, and a focus view like the traffic light, are critical for scene understanding and safe driving.

**Safe and interpretable driving** Safety has long been studied in the traditional planning/control community. However, in autonomous driving, uncertain behaviors [28, 29, 30], diverse driving preferences of drivers [31], numerous driving situations [32] deteriorate the safety concern. Traditional rule-based methods usually hand-crafted different delicate rules to tackle different driving situations [32]. However, such hand-crafted designs usually require heavy human engineering effort and it is hard to enumerate on all possible cases. In comparison, learning-based methods aims at learning diverse driving behaviors from data without heavy human design labor. However, their lack of interpretability becomes a new puzzle in the way. There are efforts taking a bypass, verifying the functioning conditions of neural network models instead of directly understanding them [9, 10, 11]. However, feedback on the failure causes and solutions are still wanted. Some works design auxiliary tasks to output interpretable semantic features [33, 34, 35], which is showing a great improvement on both the performance and interpretability. In this paper, we output the intermediate interpretable features for more semantics, which are then used in downstream controller to explicitly enhance driving safety.

## 3  Method

As illustrated in Fig. 2, the proposed framework consists of three main components: 1) a multi-view multi-modal fusion transformer encoder that integrates the signals from multiple RGB cameras and LiDAR; 2) a transformer decoder generating the low-level actions, and interpretable intermediate features including the ego vehicle's future trajectory, the object density map, and the traffic rule signals; 3) a safety controller which utilizes the interpretable intermediate features to constrain the

low-level control within the safe set. This section will introduce the input/output representation, model architecture, and safety controller.

## 3.1 Input and Output Representations

**Input representations** Four sensors are utilized, including three RGB cameras (left, front and right) and one LiDAR sensor. Four image inputs are obtained from the three cameras. In addition to the left, front and right image input $\{\mathbf{I}_{\text{left}}, \mathbf{I}_{\text{front}}, \mathbf{I}_{\text{right}}\}$, we also design a focusing-view image input $\mathbf{I}_{\text{focus}}$ to capture the status of distant traffic lights by cropping the center patch of the raw front RGB image. For the LiDAR point clouds, we follow previous works [36, 37, 8] to convert the LiDAR point cloud data into a 3-bin histogram over a 3-dimensional Bird's Eye View (BEV) grid, resulting in a two-channel LiDAR bird's-eye view projection image input $\mathbf{I}_{\text{lidar}}$.

**Output representations** Our model generates two types of outputs: safety-insensitive and safety-sensitive outputs. For safety-insensitive outputs, InterFuser predicts a path with $L = 10$ waypoints for the ego vehicle to steer toward. It guides the future driving route of the ego vehicle. However, driving along the path without a proper speed might be unsafe and violate actual traffic rules. Therefore, InterFuser additionally predicts the safety-sensitive outputs, consisting of an object density map and traffic rule information. The object density map $M \in \mathbb{R}^{R \times R \times 7}$ provides 7 features for potential objects in each grid cell, such as vehicles, pedestrians and bicycles. $M_{i,j}$ indicates an $1m \times 1m$ grid area indexed by spatial coordinates $(i, j)$ where the ego vehicle is taken as the origin and the y-axis is the forward direction. So the map covers $R$ meters in front of the ego vehicle and $\frac{R}{2}$ meters on its two sides. The 7 channels in each grid cell are the probability of the existence of an object, 2-dimensional offset from the center of the $1m \times 1m$ grid, the size of the object bounding box, the object heading, and the object velocity. Besides, InterFuser also outputs the traffic rule information, including the traffic light status, whether there is a stop sign ahead, and whether the ego vehicle is at an intersection.

## 3.2 Model architecture

**Backbone** For each image input and LiDAR input $I \in \mathbb{R}^{3 \times H_0 \times W_0}$, we use a conventional CNN backbone ResNet [38] to generate a lower-resolution feature map $\mathbf{f} \in \mathbb{R}^{C \times H \times W}$. We set $C = 2048$ and $(H, W) = (\frac{H_0}{32}, \frac{W_0}{32})$ in experiments.

**Transformer encoder** For the feature map $f$ of each sensor input, we first take a $1 \times 1$ convolution to obtain a lower-channel feature map $\mathbf{z} \in \mathbb{R}^{d \times H \times W}$. The spatial dimension of each feature map $\mathbf{z}$ is then collapsed into one dimension, resulting in $d \times HW$ tokens. Fixed 2D sinusoidal positional encoding $\mathbf{e} \in \mathbb{R}^{d \times HW}$ is added to each token to retain positional information in each sensor input, and learnable sensor embedding $\mathbf{s} \in \mathbb{R}^{d \times N}$ is added to distinguish tokens from $N$ different sensors:

$$\mathbf{v}_i^{(x,y)} = \mathbf{z}_i^{(x,y)} + \mathbf{s}_i + \mathbf{e}^{(x,y)} \tag{1}$$

Where $\mathbf{z}_i$ represents the tokens extracted from $i\text{-}th$ view, $x, y$ denotes the token's coordinate index in that sensor. Finally we concatenate the tokens from all sensors, which are then passed through a transformer encoder with $K$ standard transformer layers. Each layer $\mathcal{K}$ consists of Multi-Headed Self-Attention [7], MLP blocks and layer normalisation [39].

**Transformer decoder** The decoder follows standard transformer architecture, transforming some query embeddings of size $d$ using $K$ layers of multi-headed self-attention mechanisms. Three types of queries are designed: $L$ waypoints queries, $R^2$ density map queries and one traffic rule query. In each decoder layer, we employ these queries to inquire about the spatial information from the multi-modal multi-view features via the attention mechanism. Since the transformer decoder is also permutation-invariant, the above query embeddings are the same for the decoder and can not produce different results. To this end, we add learnable positional embeddings to these query embeddings. The results of these queries are then independently decoded into $L$ waypoints, one density map and the traffic rule by the following prediction headers.

**Prediction headers** The transformer decoder is followed by three parallel prediction modules to predict the waypoints, the object density map and the traffic rule respectively. For waypoints prediction, following [40, 10], we take a single layer GRU [41] to auto-regressively predict a sequence of $L$ future waypoints $\{\mathbf{w}_l\}_{l=1}^{L}$. The GRU predicts the $t\text{-}th$ waypoint by taking in the hidden state from step $t - 1$ and the $t\text{-}th$ waypoint embedding from the transformer decoder. Note

that we first predict each step's differential displacement and then recover the exact position by accumulation. To inform the waypoints predictor of the ego vehicle's goal location, we initialize the initial hidden state of GRU with a 64-dimensional vector embedded by the GPS coordinates of the goal location with a linear projection layer. For the density map prediction, the corresponding $R^2$ $d$-dimensional embeddings from the transformer decoder are passed through a 3-layer MLP with a ReLU activation function to get a $R^2 \times 7$ feature map. We then reshape it into $\mathbf{M} \in \mathbb{R}^{R \times R \times 7}$ to obtain the object density map. For traffic rule prediction, the corresponding embedding from the transformer decoder is passed through a single linear layer to predict the state of traffic light ahead, whether there is a stop sign ahead, and whether the ego vehicle is at an intersection.

**Loss Function** The loss function is designed to encourage predicting the desired waypoints ($\mathcal{L}_{pt}$), object density map ($\mathcal{L}_{map}$), and traffic information ($\mathcal{L}_{tf}$):

$$\mathcal{L} = \lambda_{pt}\mathcal{L}_{\text{pt}} + \lambda_{map}\mathcal{L}_{map} + \lambda_{tf}\mathcal{L}_{tf}, \tag{2}$$

where $\lambda$ balances the three loss terms. Readers can refer to Appendix A for detailed description.

## 3.3 Safety Controller Leveraging Intermediate Interpretable Features

With the waypoints and intermediate interpretable features (object density map and traffic rule) output from the transformer decoder, we are able to constrain the actions into the safe set. Specifically, we use PID controllers to get two low-level actions: the lateral steering action and the longitudinal acceleration action. The lateral steering action aligns the ego vehicle to the desired heading $\psi_d$, which is simply the average heading of the first two waypoints ahead of ego vehicle. The longitudinal acceleration action aims to catch the desired speed $v_d$. The determination of $v_d$ needs to take surrounding objects into account to ensure safety, for which we resort to the object density map.

The object in a grid of the object density map $M \in \mathbb{R}^{R \times R \times 7}$ is described by an object existence probability, a 2-dimensional offset from the grid center, a 2-dimensional bounding box and a heading angle. We recognize the existence of an object in a grid once one of the following criteria is met: 1) if the object's existence probability in the grid is higher than threshold$_1$. This criterion is intuitive. However, when an object has high position uncertainty, the probability in each grid can be not high enough to reach the threshold$_1$, leading to failure in detecting these objects. One workaround is to simply set a low threshold$_1$. However, for objects with high uncertainty, the probability in multiple grids can be similar. Thus a low threshold$_1$ may lead to multiple repeated detections for one exact object. Consequently, following [42, 43], we designed the second criteria: 2) if the object existence probability in the grid is the local maximum in surrounding grids and greater than threshold$_2$ (threshold$_2$ < threshold$_1$). This criterion can detect uncertain objects by a lower threshold$_2$, and avoid repeated detection by only picking the single grid with the highest probability. In addition to the current state of objects, the safety controller also needs to consider their future trajectory. Thus we first design a tracker to monitor and record their historical dynamics, and then predict their future trajectory by propagating their historical dynamics forward in time with moving average.

With the recovered surrounding scene and future forecasting of these objects, we then can obtain $s_t$, the maximum safe distance the ego-vehicle can drive at time step $t$. The desired velocity $v_d$ with enhanced safety is then derived by solving a linear programming problem. Note that to avoid attractions of unsafe sets and future safety intractability, we also augment the shape of objects, and consider the actuation limit and ego vehicle's dynamic constraint. For details of desired velocity $v_d$ optimization, please refer to Appendix B. In addition to the object density map, the predicted traffic rule is also utilized for safe driving. The ego-vehicle will perform an emergency stop if the traffic light is not green or there is a a stop sign ahead.

Note that while more advanced trajectory prediction methods [44, 45, 46] and safety controller [47, 48] can be used, we empirically found our dynamics propagation with moving average and linear programming controller sufficient. In case of more complicated driving tasks, those advanced algorithms can be easily integrated into our framework in the future.

# 4 Experiments

## 4.1 Experiment Setup

We implemented our approach on the open-source CARLA simulator with version 0.9.10.1, including 8 towns and 21 kinds of weather. Please refer to Appendix C for implementation details.

**Dataset collection** We ran a rule-based expert agent on all 8 kinds of towns and weather with 2 FPS, to collect an expert dataset of 3M frames (410 hours) for training and evaluation. Note that the expert agent has the access to the privileged information in the CARLA simulator. For the diversity of the dataset, we randomly generated different routes, spawn dynamic objects and adversarial scenarios.

**Benchmark** We evaluated our methods on three benchmarks: CARLA public leaderboard [49], the Town05 benchmark [8] and CARLA 42 Routes benchmark [17]. In these benchmarks, the ego vehicle is required to navigate along predefined routes without collision or violating traffic rules in existence of adversarial events[3]. At each run, the benchmark randomly spawns start/destination points, and generates a sequence of sparse goal locations in GPS coordinates. Our method uses these sparse goal locations to guide the driving without manually setting discrete navigational commands (go straight, lane changing, turning). Please refer to Appendix D for detailed descriptions of the three benchmarks.

**Metrics** Three metrics introduced by the CARLA LeaderBoard are used for evaluation: the route completion ratio (RC), infraction score (IS), and the driving score (DS). Route completion is the percentage of the route completed. The infraction score is a performance discount value. When the ego-vehicle commits an infraction or violates a traffic rule, the infractions score will decay by a corresponding percentage. The driving score is the product of route completion ratio and infraction score, and thus is a more comprehensive metric to describe both driving progress and safety.

## 4.2 Comparison to the state of the art

Table 1 compares our method with top 10 state-of-the-art methods on the CARLA leaderboard [49]. TCP is an anonymous submission without reference. LAV [51] trains the driving agent with the dataset collected from all the vehicles that it observes. Transfuser [8, 52] is an imitation learning method where the agent uses transformer to fuse information from the front camera image and LiDAR information. The entries "Latent Transfuser" and "Transfuser+" are variants of Transfuser. GRIAD [53] proposes to combine benefits from both exploration and expert data. Rails [55] uses a tabular dynamic-programming evaluation to compute action-values. IARL [6] is a method based on a model-free reinforcement-learning. NEAT [17] proposes neural attention fields which enables the reasoning for end-to-end imitation learning.

| Rank | Method | Driving Score | Route Completion | Infraction Score |
|------|--------|--------------|------------------|------------------|
| 1 | InterFuser (ours) | **76.18** | 88.23 | 0.84 |
| 2 | TCP [50] | 75.14 | 85.63 | **0.87** |
| 3 | LAV [51] | 61.85 | **94.46** | 0.64 |
| 4 | TransFuser [52] | 61.18 | 86.69 | 0.71 |
| 5 | Latent TransFuser [52] | 45.20 | 66.31 | 0.72 |
| 6 | GRIAD [53] | 36.79 | 61.85 | 0.60 |
| 7 | TransFuser+ [54] | 34.58 | 69.84 | 0.56 |
| 8 | Rails [55] | 31.37 | 57.65 | 0.56 |
| 9 | IARL [6] | 24.98 | 46.97 | 0.52 |
| 10 | NEAT [17] | 21.83 | 41.71 | 0.65 |

Table 1: Performance comparison on the public CARLA leaderboard [49] (accessed Jun 2022). All three metrics are higher the better. Our Interfuser ranks first on the leaderboard, with the highest driving score, the second highest route completion, and the second highest infraction score.

Our InterFuser ranks first on the leaderboard, with the highest driving score 76.18, the second highest route completion 88.23, and the second highest infraction score 0.84. We also compared our method with other methods on the Town05 benchmark and CARLA 42 Routes benchmark. As shown in Table 4 and Table 5 of the appendix, our method also beats other methods on these two benchmarks.

## 4.3 Ablation study

On the Town05 Long benchmark, we investigated the influence of different sensor inputs, sensor/position embedding, sensor fusion approach, and safety controller. In addition to three metrics, we also present infraction details for comprehensive analysis.

---

[3]The adversarial events include bad road conditions, front vehicle's sudden brake, unexpected entities rushing into the road from occluded regions, vehicles running a red traffic light, etc. Please refer to https://leaderboard.carla.org/scenarios/ for detailed descriptions of adversarial events.

| | Driving score | Route compl | Infrac. score. | Collision static | Collision pedestrian | Collision vehicle | Red light infraction | Agent blocked |
|---|---|---|---|---|---|---|---|---|
| | %, ↑ | %, ↑ | %, ↑ | #/Km, ↓ | #/Km, ↓ | #/Km, ↓ | #/Km, ↓ | #/Km, ↓ |
| $F$ | $40.3 \pm 4.3$ | $\mathbf{99.9} \pm 0.0$ | $40.3 \pm 5.2$ | $0.09 \pm 0.01$ | $0.04 \pm 0.01$ | $0.11 \pm 0.02$ | $0.09 \pm 0.02$ | $\mathbf{0} \pm 0$ |
| $F + Li$ | $47.2 \pm 0.9$ | $96.6 \pm 1.1$ | $48.9 \pm 2.9$ | $0.06 \pm 0.01$ | $0.02 \pm 0.02$ | $\mathbf{0.09} \pm 0.02$ | $0.10 \pm 0.02$ | $\mathbf{0} \pm 0$ |
| $F + LR$ | $43.0 \pm 5.4$ | $98.0 \pm 2.7$ | $43.2 \pm 6.2$ | $0.05 \pm 0.01$ | $0.01 \pm 0$ | $\mathbf{0.09} \pm 0.03$ | $0.12 \pm 0.03$ | $\mathbf{0} \pm 0$ |
| $F + LR + Fc$ | $46.4 \pm 3.1$ | $97.1 \pm 5.0$ | $48.5 \pm 3.3$ | $0.07 \pm 0.01$ | $0.01 \pm 0.01$ | $0.1 \pm 0.02$ | $0.04 \pm 0.02$ | $0.02 \pm 0.02$ |
| $F + LR + Li$ | $49.2 \pm 0.7$ | $89.6 \pm 0.7$ | $55.3 \pm 2.8$ | $\mathbf{0.01} \pm 0.01$ | $\mathbf{0} \pm 0$ | $\mathbf{0.09} \pm 0.05$ | $0.10 \pm 0.02$ | $0.04 \pm 0.02$ |
| $F + LR + Fc + Li$ (Ours) | $\mathbf{51.6} \pm 3.4$ | $88.9 \pm 2.5$ | $\mathbf{58.6} \pm 5.2$ | $\mathbf{0.01} \pm 0.01$ | $\mathbf{0} \pm 0$ | $\mathbf{0.09} \pm 0.05$ | $\mathbf{0.02} \pm 0.02$ | $0.07 \pm 0.01$ |

Table 2: Ablation study for different sensor inputs. $F$, $LR$, $Fc$, $Li$ denotes the front view, left and right view, focusing view, and LiDAR BEV representations respectively. ↑ and ↓ respectively denote that higher/lower metric value represents better performance. Our method performs the best when all sensor inputs are leveraged.

| | Driving score | Route compl. | Infrac. score | Collision static | Collision pedestrian | Collision vehicle | Red light infraction | Agent blocked |
|---|---|---|---|---|---|---|---|---|
| | %, ↑ | %, ↑ | %, ↑ | #/Km, ↓ | #/Km, ↓ | #/Km, ↓ | #/Km, ↓ | #/Km, ↓ |
| No sensr/pos embd | $48.6 \pm 4.7$ | $91.7 \pm 2.4$ | $52.6 \pm 6.1$ | $0.05 \pm 0.04$ | $0.01 \pm 0.01$ | $0.11 \pm 0.10$ | $0.05 \pm 0.01$ | $0.08 \pm 0.1$ |
| Concatenated input | $21.5 \pm 7.6$ | $\mathbf{97.2} \pm 3.9$ | $21.5 \pm 8.6$ | $0.06 \pm 0$ | $0.03 \pm 0.02$ | $0.26 \pm 0.05$ | $0.20 \pm 0.02$ | $\mathbf{0} \pm 0.01$ |
| No attn btw views | $46.8 \pm 6.6$ | $94.1 \pm 3.5$ | $49.6 \pm 9.8$ | $0.07 \pm 0.01$ | $0.01 \pm 0.01$ | $0.1 \pm 0.02$ | $0.04 \pm 0.02$ | $0.01 \pm 0.01$ |
| No safety contrl | $43.9 \pm 4.7$ | $96.3 \pm 5.2$ | $45.6 \pm 5.9$ | $0.05 \pm 0.03$ | $0.02 \pm 0.01$ | $0.13 \pm 0.10$ | $0.10 \pm 0.02$ | $0.08 \pm 0.2$ |
| InterFuser (ours) | $\mathbf{51.6} \pm 3.4$ | $88.9 \pm 2.5$ | $\mathbf{58.6} \pm 5.2$ | $\mathbf{0.01} \pm 0.01$ | $\mathbf{0} \pm 0$ | $\mathbf{0.09} \pm 0.05$ | $\mathbf{0.02} \pm 0.02$ | $0.07 \pm 0.01$ |

Table 3: Ablation study for sensor/position embedding (No sensr/pos embd), sensor fusion approach (Concatenated input, No attn btw views), and safety controller (No safety-contrl). The results demonstrated the effectiveness of these modules.

**Sensor inputs** As in Table 2, we evaluated the performance when different combinations of sensor inputs are utilized. In the benchmark, adversarial events such as emerging pedestrians or bikes from occluded regions are very common. Since the baseline $F$ only uses the front RGB image input, it is hard to notice obstacles on the sides of the ego vehicle. Therefore, though the baseline $F$ achieved a high route completion ratio, it significantly downgraded in the driving score and infraction score. When the left and right cameras are added, $F + LR$ achieved a higher driving score and infraction score with effectively reduced collision rate. Since traffic lights are located on the opposite side of the intersection, they are usually too far to be detectable in the original front image. Adding a focusing view for distant sight, $F + LR + Fc$ reduced the red light infraction rate by 80% compared to $F + LR$, resulting in a higher driving score and infraction score. Equipped with another Lidar input for additional 3D context, our Interfuser $F + LR + Fc + Li$ further reduced the collision rate and red light infractions, leading to the highest driving score and infraction score .

**Sensor Embedding and Positional Encoding** In InterFuser, we added the sensor embedding and the positional encoding to help the transformer distinguish tokens in different sensors and different positions in one sensor. As in Table 3, when these operations are removed, the ablation "No sensr/pos embd" had the driving score and the infraction score dropped by 3% and 6% respectively.

**Sensor Fusion Methods** In Table 3, we also evaluated the performance when different sensor fusion methods are applied. There are two common methods to fuse multi-view or multi-modal inputs: 1) directly concatenating three RGB views along the channel dimension, as in LBC [5] and IA [6]. This ablation "Concatenated input" is assumed to insufficiently fuse information and turned out to have the driving score and infraction score lower by 58% and 63%. 2) multi-stage fusion of images and LiDAR by transformer as in TransFuser [8]. Though the attention mechanism is applied, this method does not consider multi-view fusion and can have problems processing multi-view multi-modal inputs. We implemented this method by designing a mask to disallow attention among different views in the transformer encoder, resulting in drops of 9% and 15% on the driving score and the infraction score.

**Safety-enhanced controller** As in Table 3, when the safety controller is removed and our model directly outputs waypoints, the driving score and the infraction score dropped significantly by 15%. and 22%. Besides, different driving preferences can be generated by assigning different safety factors in the safety controller. As in Fig. 4 of Appendix B, the agents behaves more conservatively with a higher safety factor, leading to higher driving score and infraction score, and lower route completion.

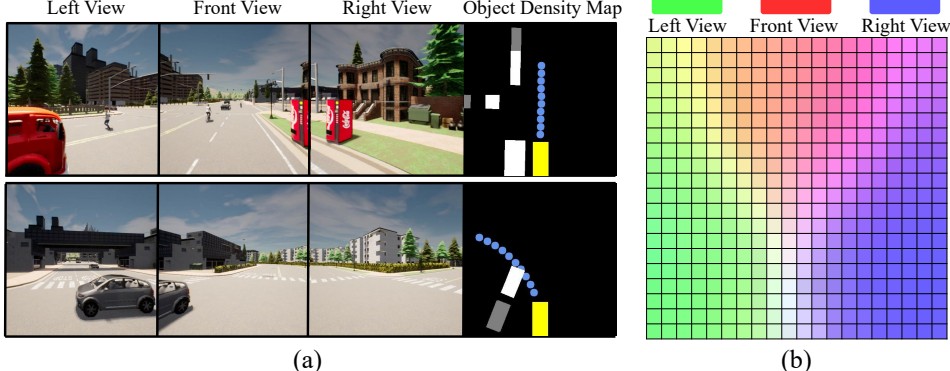

Figure 3: (a) Two cases of how our method predicts waypoints and recovers the traffic scene. Blue points denote predicted waypoints. The yellow rectangle represents the ego vehicle, and white/grey rectangles denote the current/future positions of detected objects. (b) Visualization of attention weights the between object density map queries and the features from different views.

### 4.4 Visualization

In Fig. 3 (a), we visualized two cases to show how our method predicts the waypoints and recovers the surrounding scene. In Fig. 3 (b), we also visualize the attention weights on the $20 \times 20$ object density map in the transformer decoder, to show how our method aggregates information from multiple views. As in Fig. 3 (b), our model extracts features from different views for different grids of the object density map queries. For example, the queries on the left of the map mostly attend to the features from the left views. More case visualizations can be found in Fig. 5 and Fig. 6 in the Appendix.

## 5 Limitation

Though currently ranking the first on public CARLA Leaderboard, our method still has traffic infractions. Visualization of good cases is provided in Fig. 3 and Fig. 5 of the appendix, and some failure cases are provided in Fig. 7 in the appendix. Statistical analysis of the failure causes can be found in Appendix E. Currently, we used two-threshold criteria for detection, and dynamic propagation with moving average for trajectory prediction. In the future, more advanced detection and prediction models can be used to further improve the performance. Extending the current deterministic form to a probabilistic form can also help to deal with the uncertainty and multi-modality in detection and prediction. Rigorous theoretical proof on the safety and interpretability is also an important future work. Besides, all this work happened in simulation, where driving scenarios are limited. On actual roads where countless driving situations exist, enhancing the generalizability of our method would be vital for scalable deployment.

## 6 Conclusion

We present InterFuser, a novel autonomous driving framework based on an interpretable sensor fusion transformer. The driving agent obtains a perception of global context from multi-view multi-modal inputs and generates interpretable intermediate outputs. The interpretable outputs are utilized by a safety-enhanced controller to constrain actions within the safe set. InterFuser sets the new performance upper-bound in complex and adversarial urban scenarios on the CARLA. Our future work includes importing temporal information into the pipeline for more accurate and stable prediction of other objects' future trajectories. Given that our method is flexible and extendable, it would also be interesting to explore different types of interpretable intermediate outputs, or use more advanced controllers.

## Acknowledgment

The authors would like to thank Chuming Li for insightful discussions. The work was supported by the National Key R&D Program of China under Grant 2021ZD0201300.

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
