# OpenReview forum: "Safety-Enhanced Autonomous Driving Using Interpretable Sensor Fusion Transformer"
_robot-learning.org/CoRL/2022/Conference — CoRL 2022 Poster_

### Official Review · Reviewer_snM2 · 2022-06-28

**Originality:** Good
**Technical Quality:** Fair
**Clarity Of Presentation:** Very Good
**Impact:** 3

**Recommendation:**

Weak Reject: I recommend rejecting the paper, but will not argue for my recommendation if the majority of other reviewers have a different opinion.

**Summary:**

This paper proposed a transformer fusion architecture for controlling autonomous driving agents. Various image and LiDAR inputs are processed by CNNs and fused by a transformer encoder, which is followed by a transformer decoder to output driving action and auxiliary outputs. Evaluation results on various benchmarks, including the public CARLA leaderboard, demonstrate the effectiveness of the proposal.

**Issues:**

See concerns in the weakness section.

**Quality Of The Limitations Section:**

Limitations are addressed clearly

**Reviewer Expertise:**

3: The reviewer is fairly confident that the evaluation is correct

**Robotics Focus:**

Highly relevant to robotics but no hardware experiments

**Strengths And Weaknesses:**

Strengths:

1. This paper covers an important topic.

2. The proposal achieves impressive performance.

3. The writing is generally clear and the diagrams are helpful for understanding the architecture.

Weaknesses:

My main concern about this paper is the lack of more careful analysis of safety and interpretability, the two main benefits claimed by this paper. These two concepts are very related in this paper, as safety is ensured by the "interpretable" intermediate outputs generated by the model, such as inferred traffic state information.

However, since these outputs are generated by equally non-interpretable black box model (actually the same transformer decoder), so this set up seems more like an auxilliary loss setup, rather than providing any kind of interpretability or safety guarantee.

For safety, it is mainly tackled by verification, as already discussed in the related work, or barrier functions [e.g. 1]. However, the safety notion strongly depends on the quality of the intermediate output prediction, which seems hard to offer any guarantee.

On the interpretability side, I wouldn't call such an architecture interpretable. Specifically, it is not clear what additional benefits the "explanation" (i.e. intermediate outputs) offer, in terms of understanding the model. Because the explanation is generated along side the action predictions, they do not need to be coupled in anyway, so that, for example, the action could be the "drive forward" prediction even if a red traffic light is also predicted. For claims on interpretability, I would like to see some concrete evidence, such as helping with model debugging [2, 3], improving human-model collaboration [4], or some other use cases [5].

At the very least, some more careful analysis of the intermediate output is needed to understand when they can help and when they cannot. Then the lack of rigorous studies of safety and interpretability could be acknowledged in the limitation section.

[1] https://arxiv.org/abs/2109.06689

[2] https://arxiv.org/abs/2104.14403

[3] https://openreview.net/forum?id=xNOVfCCvDpM

[4] https://arxiv.org/abs/2006.14779

[5] https://dl.acm.org/doi/10.1145/3511299

**Summary Of Recommendation:**

I believe that this paper could make an important contribution, but at this moment, neither of the central claims, safety and interpretability, are supported by strong empirical evidence. However, if such evidence is convincingly demonstrated, this paper could have potentially high impact.

---

> ### Author Response · Authors · 2022-08-22
> **Author Response to Reviewer snM2 (Part 1/2)**
>
> We appreciate the constructive comments provided and sincerely thank you for comprehensive understanding of this work from multiple aspects. We appreciate that you think highly of the importance of the topic, our performance, writing, and presentation! The following are responses to the concerns you have. We hope that they could solve your concerns on safety and interpretability. Please let us know if you want to know anything further or discuss any questions and concerns further.
>
> ---
>
> > However, since these outputs are generated by equally non-interpretable black box model (actually the same transformer decoder), so this set up seems more like an auxiliary loss setup, rather than providing any kind of interpretability or safety guarantee.
>
> **Response:** Thank the reviewer for the question. We want to clarify that the generated intermediate features are not just auxiliary outputs without further use as in [1][2]. Instead, our method directly leverages these features in the downstream safety controller. The safety controller will take these intermediate features as safety constraints and bound actions within the safe set to enhance safety. As a result, these intermediate features can provide evidence of how the model perceives its surroundings and how the decision is made, which provides more interpretability.
>
>
> ---
>
> > For safety, it is mainly tackled by verification, as already discussed in the related work, or barrier functions [e.g. 1]. However, the safety notion strongly depends on the quality of the intermediate output prediction, which seems hard to offer any guarantee.
>
> **Response:** Thank the reviewer for this question. First, we need to clarify that we claimed safety **enhancement** instead of the safety **guarantee**. In our work, the alleviate the safety issues by 1) using the sensor fusion transformer to fuse information from different sensors for better scene understanding and adversarial events detection; 2) using a safety controller to bound actions within the safe set. We agree with the reviewer that we did not achieve safety “guarantee”, thus we also fairly reported some failure cases in the limitation section and Figure 7 of the appendix. But, the extensive experiments and ablation studies on the leaderboard and benchmarks have shown that our method has a good performance and enhanced driving safety. This is why we claim the safety enhancement instead of the safety guarantee. We think such a claim is proper.
>
> ---
>
> > On the interpretability side, I wouldn't call such an architecture interpretable. Specifically, it is not clear what additional benefits the "explanation" (i.e. intermediate outputs) offer, in terms of understanding the model. Because the explanation is generated along side the action predictions, they do not need to be coupled in anyway, so that, for example, the action could be the "drive forward" prediction even if a red traffic light is also predicted. For claims on interpretability, I would like to see some concrete evidence, such as helping with model debugging [2, 3], improving human-model collaboration [4], or some other use cases [5].
>
> **Response:** Thank the reviewer for this question. There are indeed some works that output some intermediate auxiliary features along with the main actions [1]. In such cases, we agree with the reviewer that these perception features are just auxiliary outputs. These intermediate features are not used in the decision making process and would not necessarily be consistent with the final action, such as the example of running red lights mentioned by the reviewer. However, in our method, these intermediate features are actually used by the downstream safety controller. The safety controller will take these intermediate features as safety constraints and bound actions within the safe set, as shown in the right column of Figure 2. For example, when a red light is predicted, such information will be used in the controller, resulting in braking actions. As a result, these features can indeed display the actual decision basis and process, which provides more interpretability.
>
> In practice, such interpretability does help us debug and improve our model. For example, when traffic infractions occur such as collision or running a red light shown in Appendix (Figure 7), these intermediate features can tell us which module, the perception or the control, is responsible for this failure. With such clear failure causes, we can then devote to debugging and improving the specific failing module to quickly update our model. This is hard for the pure end-to-end methods [3][4] to achieve, where we have nothing else but the final actions .

---

> > ### Author Response · Authors · 2022-08-22
> > **Author Response to Reviewer snM2 (Part 2/2)**
> >
> > > At the very least, some more careful analysis of the intermediate output is needed to understand when they can help and when they cannot. Then the lack of rigorous studies of safety and interpretability could be acknowledged in the limitation section.
> >
> > **Response:** Thank the reviewer for this question. In our submitted manuscript, we do have provided some good cases (section 4.4, figure 3, and figure 5 of the appendix), and bad cases (Figure 7 of the appendix). But it seems that we did not explicitly mention these contents in the limitation section. We have added explicit references of these contents in the limitation section and thank the reviewer for helping us make our manuscript more clear.
> >
> > Additionally, we collected failing cases and analyzed the failing conditions and causes. As a detailed statistics, 45% percent of the failing cases are due to failing in detecting objects (vehicles, bicycles, etc.); 15% percent of the failing cases are due to inaccurate detection results (speed, heading, etc); 15% percent of the failing cases are due to misrecognition of the traffic lights. We have added these contents in the Appendix.
> >
> >
> >
> >
> >
> > ---
> >
> > **Reference**
> >
> >
> > [1] Chitta, Kashyap, Aditya Prakash, and Andreas Geiger. "Neat: Neural attention fields for end-to-end autonomous driving." Proceedings of the IEEE/CVF International Conference on Computer Vision. 2021.
> >
> > [2] Chitta, Kashyap, et al. "TransFuser: Imitation with Transformer-Based Sensor Fusion for Autonomous Driving." arXiv preprint arXiv:2205.15997 (2022).
> >
> > [3] Toromanoff, Marin, Emilie Wirbel, and Fabien Moutarde. "End-to-end model-free reinforcement learning for urban driving using implicit affordances." Proceedings of the IEEE/CVF conference on computer vision and pattern recognition. 2020.
> >
> > [4] Prakash, Aditya, Kashyap Chitta, and Andreas Geiger. "Multi-modal fusion transformer for end-to-end autonomous driving." Proceedings of the IEEE/CVF Conference on Computer Vision and Pattern Recognition. 2021.

---

### Official Review · Reviewer_djxH · 2022-07-29

**Originality:** Good
**Technical Quality:** Very Good
**Clarity Of Presentation:** Excellent
**Impact:** 3

**Recommendation:**

Strong Accept: I recommend accepting the paper and will argue for my recommendation even if other reviewers hold a different opinion.

**Summary:**

The authors present an interpretable autonomous vehicle (AV) policy which features a sensor fusion transformer. The authors develop a transformer encoder which takes in which uses multiple camera viewpoints and LIDAR. Additionally, a transformer decoder which output waypoints, an object density map, and traffic rules in order to determine a nominal trajectory and use a safety controller to adjust the velocity of the planned trajectory. The resulting AV policy beats state of the art methods in CARLA leaderboard and benchmarks.


**Issues:**

- Related work: The authors should include and distinguish their work from “End-To-End Interpretable Neural Motion Planner”, Zeng et al. CVPR 2019. The interpretable portion of this method is similar in concept to the one presented.

- L218: The authors do not seem to elaborate on the rule-based expert used to generate the experimental data. Is the expert based on another paper? If not, what are the rules? How important is the performance of the expert to the downstream performance of InterFuser? Would an human driver expert be preferable because of the larger variety of behaviors?

- What is the inference time for the InterFuser? Can the method be used for real-time planning/control in AV settings? If not, I suggest including this in the limitations.

- Table 2 and 3: I believe the authors have switched the route completion and infraction score column labels. If not, the interpretation of the tables 2 and 3 in L263-L288 seems incorrect. Additionally, if not, the authors need to explain why the infraction score gets worse when adding additional sensor inputs or when removing effective fusing techniques, which doesn’t make sense to me.

- Table 2 and 3: It would be helpful to include other methods to compare to in Tables 2 and 3. If other methods outperform e.g. “concatenated input” in Table 3, then it’s clear the sensor fusion portion of InterFuser is vital to success. On the other hand, if they do not outperform concatenated input, then it may be that other choices in designing the system results in the better performance.

- Appendix C: The authors should consider including a section with training time, resources used for the results in the paper, and number of parameters in the InterFuser. Since AV networks can be extremely expensive to train and evaluate, it is important to consider the accessibility of such methods which require expensive computing equipment to train.

**Quality Of The Limitations Section:**

Limitations are addressed clearly

**Reviewer Expertise:**

4: The reviewer is confident but not absolutely certain that the evaluation is correct

**Robotics Focus:**

Highly relevant to robotics but no hardware experiments

**Strengths And Weaknesses:**

Strengths:
- Justification for methods are well argued and the paper is accessible
- The results on the CARLA leaderboard and benchmarks are impressive and show the potential of the method for more realistic/larger scale settings.
- The inclusion of code for the method is a great strength and I hope it is made public upon publication. This will allow follow up work to more easily compare to this strong method and in a straightforward way and extend portions of the method (such as with different safety controllers as the authors mention on L210-L11).

Weaknesses:
- The addition of an ablation study is important in understanding the importance of the choices the authors made in the resulting method. However, as it stands, the interpretation of the ablation study and tables 2 and 3 are quite confusing and I am not sure how to view the results. The authors seem to have a different interpretation of the table than I do, but this is possibly due to a mistake in the table.
- The overall approach is very hand-crafted and there are many possibly non-obvious choices which must be made. E.g. L554-L555 cyclists’ and pedestrians’ bounding boxes are scaled up but not vehicles’. I believe this is common for large-scale learning for AVs but this does somewhat weaken the approach. It is possible that the good performance is due to these many choices and the careful cost function shaping instead of the transformer and interpretable outputs.

**Summary Of Recommendation:**

The method uses transformers in an AV sensor fusion problem and leverages them to provide useful, interpretable, intermediate features which are then used improve the safety of a planned trajectory. The results are impressive and show the strength and potential of the methods. The method seems easily adjustable to suit other problems and are amenable to follow up work. Therefore, I recommend acceptance.

---

**Update after revisions and replies**: The authors have address all major issues and weaknesses. I continue to support acceptance.

---

> ### Author Response · Authors · 2022-08-22
> **Author Response to Reviewer djxH (Part 1/3)**
>
> Thanks for the detailed and high-quality feedback. We really appreciate your recognition of the general ideas of our method, performance, and code attachment! The following are responses to the comments and suggestions you made. We hope that they can solve your concerns. Please let us know if you want to know anything further or discuss any questions and concerns further.
>
> ---
>
> > The addition of an ablation study is important in understanding the importance of the choices the authors made in the resulting method. However, as it stands, the interpretation of the ablation study and tables 2 and 3 are quite confusing and I am not sure how to view the results. The authors seem to have a different interpretation of the table than I do, but this is possibly due to a mistake in the table.
>
> **Response:** Thank the reviewer for pointing this out. We apologize that in Table 2 and 3, we mis-switched the title of two columns “infrac score” and “route compl”, which could lead to confusion. We have fixed the issues in the updated manuscripts.
> .
> ---
>
> > The overall approach is very hand-crafted and there are many possibly non-obvious choices which must be made. E.g. L554-L555 cyclists’ and pedestrians’ bounding boxes are scaled up but not vehicles’. I believe this is common for large-scale learning for AVs but this does somewhat weaken the approach. It is possible that the good performance is due to these many choices and the careful cost function shaping instead of the transformer and interpretable outputs.
>
> **Response:** We thank the reviewer for the question. There are indeed many operations in our model. However, we want to clarify that many of the operations, such as the mentioned scaling operation, are very common techniques in computer vision, which most works in the field would use as in [1][2]. For these minor choices, we did not spend much time experimenting but just chose values commonly used in most works. We explicitly wrote these techniques into the paper just for detailed description of our methods. For the reproducibility of these operations, we also displayed the exact parameter values used in our model in Table 6 of the appendix. For other major choices such as which sensors to use, we did extensive experiments and ablation studies to evaluate their effectiveness as in Table 2.
>
> As for the effectiveness of sensor fusion with transformer and safety controller with interpretable outputs, we evaluated them with extensive ablation studies as shown in Table 3. We found that using the transformer for sensor fusion can improve the driving score by 150% compared to simply concatenating sensor inputs, and the safety controller leveraging interpretable outputs can improve the driving score by 17.5%. We believe such an evaluation is fair and prove that the transformer and interpretable outputs play important roles in the good performance of our method.
>
> ---
>
> > Related work: The authors should include and distinguish their work from “End-To-End Interpretable Neural Motion Planner”, Zeng et al. CVPR 2019. The interpretable portion of this method is similar in concept to the one presented.
>
> **Response:** We thank the reviewer for raising the question.  The work “End-To-End Interpretable Neural Motion Planner” also outputs intermediate perception features for better interpretability. However, this work takes a multi-task learning approach. The perception learning is an auxiliary task and the generated intermediate perception features are not used in the downstream planning. In comparison, our method used the intermediate perception features in the downstream safety controller to bound action within safe sets to enhance safety. Thanks for the reminder and we have added the citation in the related work of our paper.

---

> > ### Author Response · Authors · 2022-08-22
> > **Author Response to Reviewer djxH (Part 2/3)**
> >
> > > L218: The authors do not seem to elaborate on the rule-based expert used to generate the experimental data. Is the expert based on another paper? If not, what are the rules? How important is the performance of the expert to the downstream performance of InterFuser? Would a human driver expert be preferable because of the larger variety of behaviors?
> >
> > **Response:** Thank the reviewer for the question. The rule-based expert agent is based on the expert agent provided in Transfuser [3]. We do find a good agent is important because it can explore longer routes and experience more scenarios, resulting in a dataset with better quality. So we did spend some time examining the agent and modifying some rules to further improve the driving performance, such as detecting the vehicles planning to change lanes. For detailed implementation please refer to the code attached.
> >
> > For the question of the human driver expert, in our simulation, a large amount of data needs to be collected from scratch. Therefore asking human experts to collect data would be too laborious and expensive to be feasible. As a result, we instead used the expert agent to collect the data. Of course, there are cases where the data is indeed available. For example, some vehicle companies would automatically collect human driver data when users drive the car. In these cases, the data is readily accessible and highly diverse, and the effort to manually design advanced expert agents is saved.
> >
> > ---
> >
> > > What is the inference time for the InterFuser? Can the method be used for real-time planning/control in AV settings? If not, I suggest including this in the limitations.
> >
> > **Response:** We thank the reviewer for the question. The inference time of Interfuser is about 0.04 second per frame on GeForce GTX 1060 (a low-end GPU), and about 0.02 second per frame on GeForce GTX 1080 Ti (a medium-end GPU). We have added this information in Appendix C of the updated manuscript. In the future, we can apply the tools of model acceleration or model quantization to further reduce the inference time. Considering that the humans’ reaction time is about 0.16 second to auditory stimulus and 0.19 second to visual stimulus [4], we believe that our method can be used for real-time planning/control in AV settings.
> >
> > ---
> >
> > > Table 2 and 3: I believe the authors have switched the route completion and infraction score column labels. If not, the interpretation of the tables 2 and 3 in L263-L288 seems incorrect. Additionally, if not, the authors need to explain why the infraction score gets worse when adding additional sensor inputs or when removing effective fusing techniques, which doesn’t make sense to me.
> >
> > **Response:** Thank the reviewer for pointing this out. We apologize that in Table 2 and 3, we mis-switched the title of two columns “infrac score” and “route compl”, which could lead to confusion. We have fixed the issue in the updated manuscripts.
> >
> > ---
> >
> > > Table 2 and 3: It would be helpful to include other methods to compare to in Tables 2 and 3. If other methods outperform e.g. “concatenated input” in Table 3, then it’s clear the sensor fusion portion of InterFuser is vital to success. On the other hand, if they do not outperform concatenated input, then it may be that other choices in designing the system results in better performance.
> >
> > **Response:** Thank the reviewer for the comments. Actually, we are a bit confused about what the ‘other methods’ refers to and how the ‘other methods’ can serve the mentioned effect. But we agree that in Table 2 and 3, we should explicitly indicate which ablation setting represents the setting of the Interfuser. We have added these notes in Table 2 and 3. Thanks for the reminder.
> >
> > Besides, here we also show an ablation study comparing our Interfuser with other sensor fusion methods and waypoint prediction methods, which further demonstrated the effect of the different parts of Interfuser.
> >
> >
> > |           Model           | Driving Score         | Route Completion              | Infraction Score        |
> > | :---------------: | :------------------: |:------------------: |:------------------: |
> > | Using ConvNet for detection    |    26.2 ± 1.2          |       70.1 ± 0.7       |    37.2 ± 3.3         |
> > | Using a direct RNN for waypoint prediction|   43.1 ± 0.7       |   75.9 ± 0.7   |   57.1 ± 1.9        |
> > | Using FC linears for waypoint prediction |    42.1 ± 0.8    |   72.5 ± 1.1    |   58.2 ± 2.3          |
> > | Using geometric-based fusion method  |   49.5 ± 0.6       |   94.1 ± 1.6      |   52.8 ± 2.2     |
> > | Using geometric-based fusion method  |   49.5 ± 0.6    |   94.1 ± 1.6     |   52.8 ± 2.2         |
> > | **Ours**  | 51.6 ± 3.4 | 88.9 ± 2.5 | 58.6 ± 5.2|

---

> > > ### Author Response · Authors · 2022-08-22
> > > **Author Response to Reviewer djxH (Part 3/3)**
> > >
> > >
> > > > Appendix C: The authors should consider including a section with training time, resources used for the results in the paper, and number of parameters in the InterFuser. Since AV networks can be extremely expensive to train and evaluate, it is important to consider the accessibility of such methods which require expensive computing equipment to train.
> > >
> > > **Response:** Thank the reviewer for pointing this out. The training time of the Interfuser is about 30 hours on 8 Tesla V100 32G graphic cards. The Interfuser consists of 52,935,567 parameters. We have added this information in Appendix C of the updated manuscript.
> > >
> > > ---
> > >
> > > **Reference**
> > >
> > > [1] Zhang, Zhejun, et al. "End-to-end urban driving by imitating a reinforcement learning coach." Proceedings of the IEEE/CVF International Conference on Computer Vision. 2021.
> > >
> > > [2] Chen, Dian, and Philipp Krähenbühl. "Learning from All Vehicles." Proceedings of the IEEE/CVF Conference on Computer Vision and Pattern Recognition. 2022.
> > >
> > > [3] Prakash, Aditya, Kashyap Chitta, and Andreas Geiger. "Multi-modal fusion transformer for end-to-end autonomous driving." Proceedings of the IEEE/CVF Conference on Computer Vision and Pattern Recognition. 2021.
> > >
> > > [4] https://en.wikipedia.org/wiki/Mental_chronometry

---

> > > > ### Comment · Reviewer_djxH · 2022-08-26
> > > > **Reply to Paper71 Authors**
> > > >
> > > > Thanks very much for the thoughtful response and the detail in addressing my comments. I appreciate the additional details on inference and training times. The elaboration provided about the expert driver and the ablation study was also helpful to clarify my understand of the paper.

---

### Official Review · Reviewer_8oVA · 2022-07-31

**Originality:** Good
**Technical Quality:** Very Good
**Clarity Of Presentation:** Very Good
**Impact:** 3

**Recommendation:**

Weak Accept: I recommend accepting the paper, but will not argue for my recommendation if the majority of other reviewers have a different opinion.

**Summary:**

This paper introduces a sensor-fusion approach that provides interpretable intermediate representations of the world scene. The approach can fuse multi-view RGB images along with LiDAR scans. The network architecture involves a CNN backbone which feeds into a transformer encoder for fusion. A transformer decoder generates an object density map, a set of waypoints for the ego, and a set of rules to be enforced (such as traffic lights) which are then fused in a safety controller to generate open-loop actions.

**Issues:**

1. Please clarify your comment about the scalability of rule-based control methods.
2. Please clarify why local max probability helps with objects suffering from high position uncertainty.
3. Please fix the grammar issues in the paper.


**Quality Of The Limitations Section:**

Limitations are addressed clearly

**Reviewer Expertise:**

3: The reviewer is fairly confident that the evaluation is correct

**Robotics Focus:**

Highly relevant to robotics but no hardware experiments

**Strengths And Weaknesses:**

Strengths:
1. I like the idea of training the perception pipeline with the planning and control modules in tow. This would allow the perception pipeline to extract the features that are relevant to behavior generation and control.
2. The approach provides interpretable perception outputs which would be a great asset for verification of the correctness of the decisions made by the downstream planning and control modules, modulo the correctness of the perception outputs.
3. The experimental results are strong (first rank in CARLA leaderboard in driving score) and thorough (extensive ablation studies that do give reasonable insights such as the importance of fusion).

Weaknesses:
1. In a simulator, the data collection for training was easy because direct access to the scene ground truth is available. However, perfectly annotating the object density map for training from real driving logs will be very challenging.
2. The planning and control modules in an actual AV might be significantly more complex than the safety controller used in this paper with multiple layers and potentially non-differentiable components.
3. The paper suffers from some grammatical errors which can be fixed.

Clarification question:
1. In the context of control, line 109 says: “rule-based methods hardly scale to complex environments due to the extensive human labor required.” What human labor is being discussed here? Also, could you cite a reference which suggests rule-based methods fail to scale. This paper (Helou, Bassam, et al. "The Reasonable Crowd: Towards evidence-based and interpretable models of driving behavior." IROS 2021) seems to suggest otherwise.
2. Why does picking the local maximum for object probability in the map help with identifying objects with high position uncertainty (as suggested in line 198)?


**Summary Of Recommendation:**

I recommend a Weak Accept. The paper is good and provides detailed experimental results. My reservations listed in weaknesses are of an “open-problems” nature.

---

> ### Author Response · Authors · 2022-08-22
> **Author Response to Reviewer 8oVA (Part 1/2)**
>
> We thank the reviewer for taking the time to review the paper and giving helpful suggestions. We are glad to know that you like our idea, approach, and experiments. We appreciate your recognition! The following are responses to comments and suggestions you made. We hope that they could solve your concerns. Please let us know if you want to know anything further or discuss any questions and concerns further.
>
> ---
>
> > In a simulator, the data collection for training was easy because direct access to the scene ground truth is available. However, perfectly annotating the object density map for training from real driving logs will be very challenging.
>
> **Response:**  We thank the reviewer for raising the question and concern. The data collection in simulators is indeed easy since the ground-truth scene information is accessible. For real driving situations, there are many open-sourced 3D detection datasets such as NuScenes, KITTI and  Waymo Open Dataset,  providing real-driving data that can be used for our training. As a result, our method can also be trained in real driving scenarios.
> Besides, we also agree with the reviewer that it is challenging to develop efficient annotation methods, and there are many research efforts devoted to self-supervised learning methods to learn without perfectly annotated data. But these are out of the scope of this paper. We admit its importance and also would like to contribute toward this direction in the future.
>
> ---
>
> > The planning and control modules in an actual AV might be significantly more complex than the safety controller used in this paper with multiple layers and potentially non-differentiable components.
>
> **Response:** We thank the reviewer for the concern. We agree that more advanced safety controllers can be used, such as [1][2]. However, in our experiments, we empirically found our simple linear-programming controller behaves decently as shown in the ablation study (Table 5). Taking another perspective, given that only very simple techniques are used, we would like to emphasize how effective the introduced multi-view-modality fusion and interpretable features are, which is the core novelty of our paper. In case of more complicated driving tasks, those advanced algorithms can be easily integrated into our framework in the future. These discussions have been included at the end of section 3.3 of the manuscripts.
>
> ---
>
> > The paper suffers from some grammatical errors which can be fixed.
>
> **Response:** We thank the reviewer for reading our paper in detail and pointing out the questions. We have proofread the paper again, polished the writing, and fixed errors.
>
> ---
>
> > In the context of control, line 109 says: “rule-based methods hardly scale to complex environments due to the extensive human labor required.” What human labor is being discussed here? Also, could you cite a reference which suggests rule-based methods fail to scale. This paper (Helou, Bassam, et al. "The Reasonable Crowd: Towards evidence-based and interpretable models of driving behavior." IROS 2021) seems to suggest otherwise.
>
> **Response:** Thank the reviewer for the question. We apologize for the unclear and inaccurate statement here which may lead to confusion. The idea we want to express is that, in autonomous driving, different driving situations require different driving and interaction patterns. For example, the driving patterns are quite different on highways (high speed and small steering) and parking lots (low speed and huge steering). The complex interaction between vehicles and human drivers also further deteriorates the problems. In such cases, rule-based methods usually need to develop different driving rules or principles for different driving cases, which requires heavy human engineer labor.  In comparison, deep-learning methods aim at learning diverse driving behaviors from data without the heavy human design labor. Such a difference of human engineer labor between rule-based methods and learning-based methods is the point we want to make. But for sure, we also think that the deep learning methods also have their issues, and the scalability in autonomous driving is still a critical problem that has not been solved. We agree that the statement on scalability failure is not accurate and we have modified the contents. Also thank the reviewer for the recommended paper, we have cited it as an example of rule-based methods testing driving performance in different cases.

---

> > ### Author Response · Authors · 2022-08-22
> > **Author Response to Reviewer 8oVA (Part 2/2)**
> >
> > > Why does picking the local maximum for object probability in the map help with identifying objects with high position uncertainty (as suggested in line 198)?
> >
> > **Response:** We thank the reviewer for raising the question. In our paper, We recognize the existence of an object in a grid by two criteria: 1) if the object's existence probability in the grid is higher than threshold$_1$. This criterion is intuitive. However, when an object has high position uncertainty, the probability in each grid can be not high enough to reach the threshold$_1$, which can lead to failure in detecting these objects. One workaround is to simply set a low threshold$_1$. However, for objects with high uncertainty, the probability in multiple grids can be similar. As a result, a low threshold$_1$ may lead to multiple repeated detections for one exact object. Consequently, we designed the second criteria: 2) if the object existence probability in the grid is the local maximum in surrounding grids and greater than threshold$_2$ ($\text{threshold}_2 < \text{threshold}_1$). This criterion can detect uncertain objects by a lower threshold$_2$, and avoid repeated detection by only picking the single grid with the highest probability. Such designs have been pretty much prevailing as in [3][4] and we have added these citations in the Section 3.3 of our paper.
> >
> > ---
> >
> > **Reference**
> >
> > [1] Leung, Karen, et al. "On infusing reachability-based safety assurance within planning frameworks for human–robot vehicle interactions." The International Journal of Robotics Research 39.10-11 (2020): 1326-1345.
> >
> > [2] Liu, Changliu, and Masayoshi Tomizuka. "Control in a safe set: Addressing safety in human-robot interactions." Dynamic Systems and Control Conference. Vol. 46209. American Society of Mechanical Engineers, 2014.
> >
> > [3]Zhou, Xingyi, Dequan Wang, and Philipp Krähenbühl. "Objects as points."
> >
> > [4]Zhou, Xingyi, Jiacheng Zhuo, and Philipp Krahenbuhl. "Bottom-up object detection by grouping extreme and center points." Proceedings of the IEEE/CVF conference on computer vision and pattern recognition. 2019.

---

> > > ### Comment · Reviewer_8oVA · 2022-08-26
> > > **Response to Authors**
> > >
> > > Thank you for answering all my questions. I have no other reservations about this paper.

---

### Author Response · Authors · 2022-08-22
**Revised Submission**

Revised PDF

---

### Meta-Review · Area_Chair_YPvk · 2022-08-15

**Recommendation:** Accept (Poster)
**Confidence:** 4

**Metareview:**

This paper proposes a new sensor-fusion approach that provides interpretable intermediate representations of the world scene and a safety-enhanced feature for autonomous driving. The authors propose to fuse multi-view RGB images along with LiDAR scans. The feature extraction part is also enhanced with the planning and control modules. The reported experiment results are promising and strong, i.e. first rank in CARLA leaderboard in driving score, and accompanied with an extensive ablation studies. The justification for the method of fusing multi-view RGB images and LiDAR scans are well articulated. The authors have greatly clarified open questions from the authors regarding the safety and interpretability. These additional details would be very helpful to understand the paper and its impact, the authors should consider to add them to the final version or its appendix.


**Best Paper Nomination:**

No

---

> ### Author Response · Authors · 2022-08-22
> **Response to Metareview**
>
> Dear Area Chair, thank you for your time, helpful comments, and recognition of our research. We have updated the manuscripts according to your and other reviewer’s suggestions and responded to each reviewer’s questions and concerns. The discussion on the safety and interpretability is mainly provided in the response to Reviewer snM2. We hope that these responses and discussions could solve those concerns. Please let us know if you want to know anything further or discuss any questions and concerns further.